# Carotenoids from Persimmon (*Diospyros kaki* Thunb.) Byproducts Exert Photoprotective, Antioxidative and Microbial Anti-Adhesive Effects on HaCaT

**DOI:** 10.3390/pharmaceutics13111898

**Published:** 2021-11-08

**Authors:** Sara Gea-Botella, Bryan Moreno-Chamba, Laura de la Casa, Julio Salazar-Bermeo, Nuria Martí, María Concepción Martínez-Madrid, Manuel Valero, Domingo Saura

**Affiliations:** 1Instituto de Investigación, Desarrollo e Innovación en Biotecnología Sanitaria de Elche (IDiBE), Universidad Miguel Hernández de Elche, 03202 Alicante, Spain; sara_gb@hotmail.es (S.G.-B.); bryan.morenoc@umh.es (B.M.-C.); lauradlad@gmail.com (L.d.l.C.); julio.salazar@goumh.umh.es (J.S.-B.); nmarti@umh.es (N.M.); dsaura@umh.es (D.S.); 2Departamento de Agroquímica y Medio Ambiente, Universidad Miguel Hernández de Elche, 03312 Alicante, Spain; c.martinez@umh.es

**Keywords:** *Diospyros kaki*, UV radiation, photoprotection, reactive oxygen species, antimicrobial activity, skin-care cosmetics

## Abstract

Persimmon (*Diospyros kaki* Thunb.) fruits are a remarkable source of carotenoids, which have shown protective effects against UV radiation in bacteria, fungi, algae, and plants. The aim of this study was to analyze the photoprotection provided by an acetone extract, rich in carotenoids and obtained from byproducts derived from the persimmon juice industry, against UV-induced cell death in the keratinocyte HaCaT cell line. For this purpose, the cytotoxicity and phototoxicity of carotenoid extract, as well as its intracellular reactive oxygen species (ROS) scavenging and anti-adhesive activities towards HaCaT cells, were evaluated. The in vitro permeation test provided information about the permeability of the carotenoid extract. Persimmon extracts, rich in carotenoids (PEC), were absorbed by HaCaT keratinocyte cells, which reduced the UV-induced intracellular ROS production in treated cells. Thus, PEC exerted a photoprotective and regenerative effect on UV-irradiated HaCaT cells, and this protection was UV dose-dependent. No cytotoxic effect was observed in HaCaT cultures at the concentration tested. PEC treatment also stimulated the adhesion capacity of skin microbiome to HaCaT cells, while exhibiting a significant anti-adhesive activity against all tested pathogens. In conclusion, PEC showed potential for use as a functional ingredient in skin-care products.

## 1. Introduction

Epithelial tissues, that conform to the skin on all human bodies, are essential for protecting human organisms from exogenous threats, such as physical, chemical, and biological pollutants [1]. Epidermis, the outermost layer, composed mainly of keratinocytes, is often the first to be affected by environmental factors, such as solar radiation, oxidants, smoke, or infective microorganisms; constant exposure to external factors may alter skin health functions, such as thermoregulation, metabolism, homeostasis, sensing, and light-dependent production of vitamin D, among others [2,3].

Ultraviolet (UV) radiation is an important exogenous factor in the pathogenesis of human skin and can lead to the development of a series of skin disorders, including sunburn (erythema and edema), hyperplasia, carcinogenesis, DNA damage, immunosuppression, and photoaging [4,5,6,7,8]. Scientists categorize UV light into three different subtypes, according to wavelength: UV-A (λ = 320–400 nm) is UV light with the longest wavelength, and the least harmful, UV-B (λ = 290–320 nm), is mostly absorbed (95%) by the ozone in Earth’s atmosphere; UV-C (λ = 100–290 nm) is extremely dangerous and is almost completely absorbed by Earth’s atmosphere [9].

The penetration efficacy of UV in human skin depends mainly on structural features and pigmentation. UV-B hardly passes beyond the epidermal layer; however, it does provoke inflammation, associated with local pain, reddening, and hyperthermia, as well as DNA damage; on the other hand, UV-A is involved in photoaging and photocarcinogenesis processes with a high reactive oxygen species (ROS) production [10,11]. The excessive generation of ROS, especially in the epidermis, results in an increase of lipid peroxidation, formation of protein adducts, apoptosis, and severe inflammatory events. Additionally, as the concentration of ROS towards deeper layers of dermal stratum increases, the outermost layers of dermis become vulnerable and more sensitive to outdoor stressors [12].

The weakening of skin shielding after solar/light radiation damage could lead to the eventual predisposition to exogenous threats, such as microbial infections. Human skin is colonized by millions of microbes that compose the skin microbiome; however, photodamage could disturb the commensals/pathogens ratio in the microbiome, leading to skin infections or even systemic diseases. For instance, it has been reported that psoriasis filamentous bacteria in skin can promote the accumulation of proinflammatory TH17 and TH1 cells, with an increase of inflammatory skin disorders [1,13].

Photodamage could also alter the skin homeostasis, with an impact on the differentiation, proliferation, and growth of cutaneous commensals and pathogens on the skin surface [13]. The changes in skin physiology site after UV exposition can cause unwanted changes by lipophilic microorganisms in sebaceous sites and the proliferation of pathogenic microbes, such as *Staphylococcus aureus* or some fungal strains [3]. Thus, an adequate supply of nutrients to the skin is required to guarantee its correct function, including photoprotection and antimicrobial shielding effects for therapeutic or preventive purposes [2]. The exogenous skin application of protective dermatological preparations containing topical sunscreens is commonly recommended [1,14,15,16].

Carotenoids are organic pigments from the group of isoprenoids that are found naturally in plants and algae, as well as several bacteria and fungi. At least part of the resistance of an Antarctic *Microbacterium* spp. against UV-B radiation relates to photoprotection by its carotenoid pigments [17]. Likewise, carotenoids and other health-promoting compounds, such as chlorophylls and flavonoids, can be elicited by increasing the levels of supplemental UV-B radiation in plants, bacteria, and algae [18,19,20,21]. Overall, natural carotenoids have evolved to play key roles in absorbing different wavelengths, according to their structure [12]. As lipophilic antioxidants, their topical application is closely related to skin protection from environmental factors, such as radiation or other external agents. Carotenoids had shown protective effects against UV radiation through their anti-inflammatory activity [1,16]. Lutein is able to protect against UV radiation-mediated skin irritation by suppressing the production of inflammatory mediators in keratinocyte (HaCaT) cells [15]. Similarly, astaxanthin inhibits inflammatory cytokine secretion from epidermal keratinocytes, in response to UV-B radiation [22]. Lycopene has been reported as the antioxidant that is most quickly depleted in skin upon UV irradiation; thus, it might play a protective role [23]. Even more, the presence of carotenoids reduces DNA damage when rat epithelial cells are exposed to UV-A radiation [24].

In addition, provitamin A carotenoids, such as α- and β-carotene and β-cryptoxanthin, are beneficial for skin, through the production of retinoic acid, which intervenes in processes including keratinocyte proliferation, epidermal differentiation and keratinization, reduction of inflammation or oxidation, enhancement of permeability of topical agents, treatment of acne, and other skin conditions [1,3]. Nowadays, the use of natural products in cosmetics for skin care photoprotection is of great interest, due to the benefits that generate these products, great acceptability by users, and low probability of systemic absorption [14]. Skin permeation of biocompounds depends upon the molecule’s physiochemical properties, such as lipophilicity, molecular weight, and shape [25]. The functional effect of a carotenoid is determined by its permeation and subcellular localization [26].

Persimmon (*Diospyros kaki* Thunb.) fruits are a remarkable source of carotenoids; moreover, studies have recently reported that the byproducts derived from persimmon juice industrial processing still hold high concentrations of these bioactive pigments [27,28]. The aim of this study was to investigate whether these byproduct extracts, high in carotenoids, showed biological activity in HaCaT cell monolayers exposed to UV-A- and UV-B-induced damage, commensal, and pathogen microorganisms from human skin and for in vitro permeation kinetics.

## 2. Materials and Methods

### 2.1. Chemicals and Reagents

The following were obtained from Sigma-Aldrich (Madrid, Spain): 3-(4,5-dimethylthiazol-2-yl)-2,5-diphenyltetrazolium bromide (MTT), 2′,7′-dichlorofluorescein diacetate (DCFH-DA), dimethyl sulfoxide (DMSO), and β-carotene. Modified Eagle’s medium (DMEM, 4.5 g/L glucose), fetal bovine serum (FBS), penicillin/streptomycin, and Dulbecco’s phosphate buffered saline (DPBS) were purchased from Gibco Laboratories (Thermo Fisher Scientific, Madrid, Spain). Strat-M^®^ Membranes were purchased by Millipore Corporation (Darmstadt, Germany). Acetone (Ac), ethanol (EtOH), and methanol (MeOH) were obtained from Panreac Química SLU (Barcelona, Spain). Ultrapure water was obtained from a purified water system Q-Gard^®^ 1 from Merck Millipore (Darmstadt, Germany), with a resistivity of 18.0 MΩ×cm.

### 2.2. Plant Material and Carotenoid-Rich Acetone Extraction

Persimmon byproduct (pulp, skin, and seeds) was provided by Mitra Sol Technologies S.L. (Elche, Spain), from the waste generated during the industrialization of persimmon juice. The persimmon byproduct was subjected to a solid–liquid extraction with acetone (1:3 *w*/*v*). The mixture was filtered to obtain liquid extract that was saponified with KOH and decanted with diethyl ether (1:2 *v/v*). Then, the pellet was washed with water to decant the resins generated in the saponification. The solvent was evaporated by a rotary evaporator under vacuum (Series R-210, Büchi, Barcelona, Spain) and lyophilized (Telstar Cryodos-80, Terrassa, Barcelona, Spain) to remove the remaining water. The carotenoid extract (oily-appearance and bright orange color) was kept under nitrogen atmosphere at −80 °C, until further use.

### 2.3. Cellular Assays

#### 2.3.1. Culture of HaCaT Cell Line

HaCaT cell line was purchased from the American Type Culture Collection (Rockville, MD, USA). Cells were cultured in DMEM, supplemented with FBS, 50 U/mL penicillin, and 50 mg/mL streptomycin, in 75 cm^2^ flasks. The culture medium was changed twice a week and was maintained at 37 °C with CO_2_ (5% *v/v*).

#### 2.3.2. Cytotoxicity Assay

Cytotoxicity assay was performed, according to [29]. Cells were seeded in 96-well plates, at density of 7 × 10^3^ cells per well, for 24 h. Considering carotenoids solubility in the culture media, cells were treated with different concentrations (10, 20, 50, 80, 110, 150, 180, and 200 µg/mL) of persimmon extract rich in carotenoids (PEC) for 48 h. After incubation time, DMEM was removed, MTT (5 mg/mL) was added, and cells were incubated at 37 °C under CO_2_ atmosphere (5% *v/v*) for 2 h. The supernatant was discarded and 100 µL of DMSO were added. Cell viability was measured at 570 nm with Spectrostar Omega (BMG Labtech GmbH, Offenburg, Germany).

#### 2.3.3. In Vitro UV Photoprotective Assay

The photoprotective activity of PEC on HaCaT cells, exposed to UV-A and UV-B irradiation, was determined according to [30]. HaCaT cells were seeded in 96-well plates (7 × 10^3^ cells per well) and incubated overnight. The pre-treated group was exposed for 3 h (short pretreatment) or 24 h (long pretreatment) at two different concentrations of PEC (20 or 50 µg/mL of DMEM per well) before UV-A and UV-B irradiation. The post-treated group was exposed with the same concentrations of PEC after UV-A and UV-B irradiation. Incubation conditions were 37 °C under CO_2_ atmosphere. For irradiation, culture media was removed, and cells were washed with PBS before exposure and kept in 50 µL of PBS during irradiation. Cells were exposed to different doses of UV-A (5 and 20 J/m^2^) and UV-B (500 a 1000 J/m^2^) irradiation through microprocessors Bio-Link^®^ Crosslinker BLX-E312 and BLX-E365 (Biotech SL, Vilber Lourmat, France), sources of UV-A and UV-B radiation, respectively. Photoprotection effect in HaCaT cell viability was determined by MTT assay.

#### 2.3.4. Intracellular ROS Scavenging Activity

DCFH-DA fluorescent probe was used to determine intracellular UV-B-induced ROS generation, in presence of PEC on HaCaT cells, according to [31], with modifications. Briefly, the cells were seeded in 96-well plates and treated with PEC (0.1, 0.5, 1, 5, 10, 25, 50, and 75 µg/mL) for 24 h (37 °C under CO_2_ atmosphere). HaCaT cells were irradiated with UV-B at 500 J/m^2^ intensity and then washed with PBS. Then, the cells were treated with DCFH-DA probe, diluted in DMEM, and further incubated for 1 h. The medium was discarded, and the cells were washed with PBS; fluorescence was measured immediately using multiplate-reader (Cytation™ 3 Cell Imaging Multi-Mode Reader, BioTek Instruments, Inc., Bad Friedrichshall, Germany). Trolox-treated cells (CT) and non-PEC-treated cells, containing DMSO (CD), were used as positive and negative controls, respectively. The assays were carried out in triplicate.

### 2.4. Microbial Assays

#### 2.4.1. Microbiology Culture Media and Reagents

Microbiological culture media used in this study consisted of potato dextrose broth (PDB) and buffered peptone water (BPW) (purchased from Scharlab (Barcelona, Spain)); tryptic soy broth (TSB) and tryptic soy agar (TSA) (from Labkem (Barcelona, Spain)); potato dextrose agar (PDA) (from Condalab (Madrid, Spain)); reinforced clostridial medium (RCM) (from Thermo Scientific (Madrid, Spain)). Antibiotics, such as kanamycin, gentamycin, erythromycin, ketoconazole, and triphenyl tetrazolium-chloride (TTC) (from Sigma-Aldrich (Madrid, Spain)), were also used on these assays.

#### 2.4.2. Microbial Strains and Inoculum Preparation

Commensal skin bacterial strains *Cutibacterium acnes* (CECT 5684) and *Staphylococcus epidermidis* (CECT 232), as well as skin pathogens, such as *Escherichia coli* (CECT 515), *Staphylococcus aureus* (CECT 59), and *Trichophyton rubrum* (CECT 2794), were obtained from the Spanish Type Culture Collection (CECT, Universitat de València, Valencia, Spain). *Candida albicans* (CLA22) was provided by Dra. Maria Francisca Colom Valiente (Departamento de Producción Vegetal y Microbiología, Facultad de Medicina, Universidad Miguel Hernández, Campus de Sant Joan D’Alacant, Alicante, Spain).

The inoculum activation of every microorganism was performed according to provider specifications. After incubation time of each microorganism, all microbial suspensions were normalized to a 0.5 McFarland standard (10^8^ colony forming units, CFU/mL) in 1% BPW, using a plate reader at 600 nm to measure optical density (OD) for bacterial count. Freshly serial dilutions were prepared in tubes with 1% BPW for assays.

#### 2.4.3. Antimicrobial Assay

Prior to antimicrobial assays, PEC was dissolved in 70% DMSO. A modified colorimetric broth microdilution method was performed to individually determine the minimum inhibitory concentration (MIC), minimum bactericidal or fungicidal concentration (MBC or MFC), and sub-inhibitory concentration (sIC) of PEC against some bacteria and fungi resident on the human skin microbiome [32]. The antimicrobial assays were carried out by triplicate.

Briefly, 100 µL of TSB (bacteria), PDB (fungi), or RCM (*C. acnes*) were added to a 96-well microtiter plate. Then, 100 µL of PEC was added in the first column of plate. Serial two-fold dilutions were made in a concentration range from 60 to 0.117 mg/mL per well, with the final aliquot being discarded. Finally, 100 µL of microbial suspension at 10^8^ CFU/mL were added to each well. Kanamycin (200 µg/mL), gentamycin (20 µg/mL), and erythromycin (8 µg/mL) were used as positive controls for bacteria. Ketoconazole (500 µg/mL) was used as positive control for fungi plates. Untreated microorganisms in their corresponding culture media were used as negative control. To confirm no effects on tested strains, DMSO (17.5%) was also tested with microorganisms at the concentration used. Blanks of culture media and PEC were incubated, as well.

The plates were incubated at 37 °C for 24 h (bacteria) and 25 °C for 3 to 5 days (fungi). *C. acnes* was incubated in anaerobic conditions. After incubation, OD was measured, and percentage of inhibition (%Inh) of the microbial population was calculated using the following equation:(1)$$\%\mathrm{Inh}=\frac{{\mathrm{OD}}_{\mathrm{control}}{-\mathrm{OD}}_{\mathrm{experimental}}}{{\mathrm{OD}}_{\mathrm{control}}}\times 100$$
where OD_control_ and OD_experimental_ were the optical density, in absorbance units read at 600 nm, for negative control and cells exposed to PEC, respectively.

To confirm inhibition data, 10 µL of aqueous 0.5% TTC solution was added to each well and further incubated for 1 h. The plates were examined to determine color change. The MIC was defined as the minimum concentration of PEC, where there was an appreciable color change, compared to %Inh (>50%). The sIC were PEC concentrations that generated %Inh between 0 to 50%. Optical images of wells containing PEC MIC-treated and untreated live cells were performed using the Cytation 3 microplate reader.

MBC or MFC were determined via the culture drop spreading method. Aliquots of 10 µL from MIC and pre-MIC wells were separately plated on TSA, PDA, or RCM medium surfaces and allowed to dry at room temperature. Plates were incubated according to each microorganism’s requirements. MBC or MFC were defined as no microbial growth observed (>99.9% killing).

#### 2.4.4. Inhibitory Microbial Adhesion to HaCaT Cell Monolayers

HaCaT cells were cultivated and grown, according to [33], with minor modifications. Briefly, 1 × 10^5^ cells per well were inoculated in 24-well plates. After monolayer formation, PEC was diluted in antibiotic-free DMEM, to reach MIC/4 (sIC), and added to every well. Then, aliquots of 500 µL of microbial suspension (10^3^ CFU/mL) were added to each well, reaching a final volume of 1 mL per well. Plates were incubated at 37 °C under humidified atmosphere of CO_2_ (5% *v/v*) for 5 h. After 1, 2, 3, 4, and 5 h of incubation, cells were washed three times with sterile PBS and then trypsinized. Aliquots of 100 µL were plated for microbial counting onto TSA, PDA, or RCM plates and incubated according to each microorganism’s requirement. Wells untreated with sIC PEC were negative control, while wells with 100 U/mL penicillin and 100 µg/mL streptomycin sulphate were positive control. To ensure sterility of the assay, a blank of wells, without microbial suspension, was assessed. Microbial adhesion curve to HaCaT cell monolayer was plotted based on the final number of adhered bacteria in untreated groups (100% adhered microbial cells) after 5 h of incubation. The assay was performed by triplicate.

### 2.5. In Vitro Permeation Test

#### 2.5.1. PEC Preparation

Due to the low solubility of carotenoids in polar solvents, 120 μg/μL PEC-DMSO or β-carotene-DMSO solutions were prepared, according to [34], and assessed for in vitro permeation in a Franz diffusion cell model.

#### 2.5.2. Franz Diffusion Cell System

The cutaneous accumulation dynamics of PEC and β-carotene, as positive control, were analyzed in vitro using Franz diffusion cells [35]. Briefly, receptor cells from Permegear^®^ Inc. (Hellertown, PA, USA) were filled with 5 mL, 1 mM phosphate buffer (pH 7) with 5% DMSO. A 9 mm diameter synthetic Strat-M^®^ membrane (EMD Millipore Corporation, Darmstadt, Germany) was placed on top, fixed with the donor compartment lid (diffusion area: 0.636 cm^2^), and left to equilibrate for 30 min in contact. The receptor cell temperature and stirring conditions were, respectively, stabilized at 32 ± 0.5 °C and 200 rpm, prior to the running of the permeation experiment; 314.3 μL/cm^2^ PEC-DMSO or β-carotene-DMSO solutions were applied to the donor cell.

Through the sampling arm, 200 µL aliquots were withdrawn at 0.5, 1.5, 2, 3, 4, 5, and 24 h from the placement of solutions; then, withdrawn aliquots were immediately replaced with fresh receptor medium. The assay was carried out in triplicate.

#### 2.5.3. High Performance Liquid Chromatography and Tandem Mass Spectrometry (HPLC-PDA-MS/MS) Analysis

Quantification and carotenoid profile of permeated PEC-DMSO solution into receptor cells were determined by HPLC-PDA-MS/MS [27]. The mobile phases and gradient conditions used for HPLC carotenoid analysis are shown in Appendix A, while Appendix A summarizes the MS parameter settings for optimized analysis, performed in a Shimadzu LCMS-QP (Kyoto, Japan), equipped with binary pumps (LC-40D X3) and connected, in series, to a photodiode array (PDA) detector (SPD-M30A) and triple quadrupole mass spectrometer (LCMS-8050TM). The UV-visible spectra were obtained between 250 and 700 nm, and the chromatograms were processed at 450 nm; 1 µL volume of each of the withdrawn aliquots of permeated PEC-DMSO solution was subjected to carotenoid analysis.

Calibration curves for lutein, β-cryptoxanthin, β-carotene, and lycopene with a minimum of five concentration (0.01–0.50 mg/mL) levels were built. Retention times of carotenoid molecular species were determined using standard compounds. UV-visible (λmax) and mass spectra were compared with data available in literature [36,37,38,39,40].

### 2.6. Statistical Analysis

The statistical analysis of obtained results was performed using GraphPad Prism 8.0.2. software (GraphPad Software, Inc., San Diego, CA, USA), with one-way analysis of variance (ANOVA) and Dunnett’s post hoc test to compare the means from several experimental groups against a control group mean to see is there is a significant difference. Tukey’s post hoc test was performed for antimicrobial activity comparison.

## 3. Results

### 3.1. Cytotoxic Activity of PEC in HaCaT Cells

MTT assay was performed to study the HaCaT cell viability against different concentrations of PEC (Figure 1). Results showed that PEC, at the concentration range between 10 and 180 µg/mL, showed no toxic effect in exposed cells; as a result, cell viability showed the same behavior as the control group (≈100% viability). When HaCaT cells were treated with 200 µg/mL of PEC, a significative decrease in exposed cells (*p* < 0.05) was observed, compared with the negative control. Given that PEC showed low cytotoxic effects, even at the highest concentration tested; concentrations higher than 200 µg/mL could also be feasible to apply. However, the solubility properties of carotenoids must be considered.

### 3.2. Photoprotective Activity of PEC against UV-A and UV-B Irradiation

UV irradiation intensities were selected in a preliminary study, in which seven different intensities of UV-A (1, 5, 8, 10, 15, 20, and 25 J/m^2^) and five of UV-B (250, 500, 750, 1000, and 1250 J/m^2^) were tested. For UV-A irradiation, the intensity of 5 J/m^2^ was chosen for further assays, since it showed over 50% of cell survival, in comparison with non-irradiated control; whereas the intensity of 20 J/m^2^ was also chosen because it showed low cell viability, about 20%. For UV-B irradiation, the intensities of 500 J/m^2^ and 1000 J/m^2^ were chosen for further assays, according to the same criteria. Figure 2 shows the viability of pre- and post-treated HaCaT cells with low (lPEC, 20 µg/mL) and high (hPEC, 50 µg/mL) concentrations of PEC, exposed to different radiation intensities of UV-A (5 and 20 J/m^2^) or UV-B (500 and 1000 J/m^2^).

As expected, a UV dose-dependent effect on cell death was observed, being more effective for higher UV intensities. Regardless of the UV radiation subtype used, the effect of PEC concentration on the viability of the irradiated HaCaT cells depended on the time at which the PEC was added. For instance, after long PEC treatment times, the presence of carotenoids in HaCaT cells slightly reduced cell viability, compared to untreated irradiated keratinocytes. On the contrary, the addition of PEC to cells after UV exposure enhanced the viability of the irradiated monolayer cultures of HaCaT keratinocytes. This positive effect was not observed when HaCaT cells were irradiated at 20 J/m^2^ UV-A intensity (Figure 2A).

This result could be due to several reasons, including the variability of both UV-irradiated HaCaT cell cultures and viability counts. Finally, after short PEC treatment times, no significant differences in cell viability were observed between treated and untreated irradiated keratinocytes. Although this described cell response was reproduced at the high intensities of UV radiation, tested in these photoprotective experiments, perhaps significantly lower doses would have made more definite and clear results possible. In any case, the overall results suggest a photoprotective and regenerating effect of PEC treatment in the monolayer cultures of HaCaT keratinocytes subjected to UV irradiation.

### 3.3. Effect of PEC on UV-B-Induced ROS Production in Treated HaCaT Cells

Intracellular ROS levels can be increased after UV-B exposure [41]. In this sense, HaCaT keratinocytes were treated with 0.1–75 µg/mL PEC concentration range for 24 h and then were irradiated with UV-B (500 J/m^2^). Intracellular ROS production was determined, in comparison with the Trolox (CT) positive control, while DMSO (CD) served as a negative control. PEC treatment at the concentration range of 0.1–5 µg/mL reduced intracellular ROS levels by a similar percentage to both cell controls (Figure 3). Nevertheless, administration of ≥10 µg/mL PEC concentrations showed a marked decrease in ROS reduction.

### 3.4. Antimicrobial Activity Results

In this study, all tested strains were susceptible to PEC (Figure 4A). Thus, PEC exhibited notable antifungal activity against *C. albicans* and *T. rubrum*, with MIC values of 3.75 and 0.94 mg/mL, respectively, as well as MFC values of 30.00 and 7.50 mg/mL. MIC values for *C. acnes*, *E. coli*, *S. aureus*, and *S. epidermidis* bacterial strains were 15.00, 7.50, 3.75, and 3.75 mg/mL, respectively. MBC value for pathogens *E. coli* and *S. aureus* were 60.00 mg/mL, while bactericidal effect of PEC against *C. acnes* and *S. epidermidis* were not detected at the concentration levels tested.

According to the used broth microdilution method to determine MICs, PEC produced higher % inhibition in fungi (67.01 ± 0.26% for *C. albicans* and 66.89 ± 1.05% for *T. rubrum*) than bacteria, apart from *E. coli* strain, which showed no significant differences (65.24 ± 1.40%) (*p* > 0.05). As expected, the remaining bacteria presented >50% Inh. Images of the MIC effect of PEC on tested microbial populations are shown in Figure 4B–G.

PEC induced a significant reduction of microorganism populations, especially compared to untreated control containing DMSO, where a dense biomass is notable after incubation period. PEC-treated wells showcased as less dense, with isolated biomasses, than the negative control.

### 3.5. Anti-Adhesive Effect of PEC on HaCaT Monolayer Cells

The effect of PEC on microbial adhesion to HaCaT monolayer cells was assessed. As seen in Figure 5, treatment with sIC (MIC/4) of PEC led to a significant decrease of *S. aureus* (*p* < 0.001) and *E. coli* (*p* < 0.001) adhesion to HaCaT keratinocytes after 5 h of co-incubation, in comparison with untreated controls, reaching percentages of adhered microorganisms equal to 13.41 ± 2.11 and 20.95 ± 3.43, respectively (Figure 5C,D). A progressive adhesion reduction of these strains to HaCaT cells could be observed after a short period of co-incubation (just 2 h).

*C. acnes* (*p* < 0.05), *S. epidermidis* (*p* > 0.05), *C. albicans* (*p* < 0.001), and *T. rubrum* (*p* < 0.01) increased their adhesion capacity to HaCaT monolayer cells during co-incubation period, in comparison with untreated controls, reaching percentages of adhered microorganisms equal to 76.10 ± 7.86, 106.67 ± 10.41, 52.53 ± 1.87, and 29.73 ± 9.36, respectively (Figure 5A,B,E,F). In these cases, PEC treatment enhanced microbial adhesion to HaCaT cells after 2–3 h of co-incubation. Results showed PEC (used at sIC) negatively affected the adhesion mechanism of pathogens *E. coli* and *S. aureus* to HaCaT monolayer cells, while stimulated adhesion capacity of the human skin common residents *C. acnes* and *S. epidermidis*. PEC-induced stimulation of microbial adhesion was significantly lower for the dermatophyte fungus *T. rubrum* and the opportunistic pathogenic yeast *C. albicans*. In general, PEC manifested a significant anti-adhesive activity against all tested pathogens (Figure 5C–F), compared to positive control (penicillin and streptomycin solution).

### 3.6. HPLC-PDA-MS/MS Analysis of Permeated Samples

To evaluate the topical release of PEC in potential transdermal therapeutics, the accumulation dynamics of its carotenoids were studied in Franz diffusion cells. Results were compared to control sample of β-carotene (Figure 6). MS/MS analysis of samples in multiple reaction monitoring (MRM) mode displayed that from the carotenoids found in PEC, (cis)-α-cryptoxanthin, (cis)-β-cryptoxanthin, (cis)-α-carotene, (cis)-β-carotene, (All-trans)-α-carotene, and (All-trans)-β-carotene, exhibited higher flux than the remaining carotenoids, as they were present in the receptor medium (Appendix A).

The cumulative amount of diffused PEC carotenoids, at a sampling time of 24 h, was 213.7 ± 2.0 μg/cm^2^, which was 34% higher than that of β-carotene (159.3 μg/cm^2^). The flux of PEC carotenoids was similar ot β-carotene over the 0.5–5 h period. A constant release pattern was observed in β-carotene over the 5–24 h period, and it is consistent with previous permeation studies [42]. The carotenoid fluxes of PEC and β-carotene were 8.8 and 6.6 (μg/cm^2^ h), respectively. PEC carotenoids showed a higher diffusion rate than β-carotene, which could be due to the diverse carotenoid composition of PEC.

## 4. Discussion

### 4.1. UV Radiation and ROS Production

European people are exposed to an average solar radiation of 10,000–20,000 J/m^2^ per year [43]. The skin acts as an external barrier in the human body against solar radiation and is constantly exposed to UV radiation, causing severe problems, such as photoaging and skin carcinomas [8,44]. Currently, one of the greatest social interests is the use of natural compounds, both in food and dermatological or cosmetic products [45]. There are many studies that focus on the development of photoprotective agents for the skin, with natural compounds that provide high antioxidant activity against exposure to UV-B radiation and ROS, both exogenous and endogenous [29].

The photoprotective experiments demonstrated that PEC, at a concentration of 50 µg/mL, exerts a regenerative effect on previously UV-B irradiated HaCaT cell monolayers. Increases in HaCaT cell viability could be due to a regulation of apoptotic markers [46], a regulatory effect on intracellular ROS production, and a reduction of DNA damage [29].

According to [47,48], UV radiation mediates its biological effects on biomembranes predominantly via the reaction of ROS on the cell membrane, leading to the deterioration of the membrane lipids, rearrangement of the phospholipid bilayer, and pore formation. Such alterations may increase membrane permeability, allowing for rapid and massive entry of PEC that would block ROS production and activate a complete system for the repair of peroxidized cell membranes [49]. In contrast, extensive membrane permeabilization by UV radiation could be detrimental to cell viability. For both PEC pretreatment experiments, the carotenoids would be either outside or on cell surface, a small part would have reached the interior of the cells during the long pretreatment, causing little inhibition of oxidative stress and null cellular response. Everything mentioned above suggests that PEC could be an ideal natural ingredient for incorporation in dermatological products, such as after-sun lotion or cream.

Particularly, the effect of PEC on ROS generated by UV-B radiation was investigated in previously irradiated HaCaT cell monolayers. PEC induced a slight, but insignificant, reduction of ROS, at both the 0.1 and 0.5 µg/mL concentrations. At high PEC concentrations (1–75 µg/mL), a decrease in ROS reduction was observed, which might be due to interference in the fluorescence measurement of the DCF-DA probe with the carotenoids. This is true because both the DCF-DA probe used for the evaluation of ROS compounds and carotenoids present in the PEC have their highest absorbance at 500 nm. It is important to note that it would have been very interesting to use another measurement probe for the ROS compounds that did not interfere with carotenoids [50].

### 4.2. Microbiology

Microbial community assembly, stability, and function are driven by host factors, as well as the interactions between these microorganisms. As in the gut, the skin microbiome can act competitively to exclude one another or synergistically for mutual benefits. Atopic antibiotic treatments could also disturb the equilibrium of the microbiome, leading to some serious skin infections [13]. Despite the availability of a variety of natural and synthetic antibiotics, the potential of carotenoids as novel antimicrobials has increased recently. Several reports have recorded the antimicrobial potential of carotenoids from seaweed [51] and fungal extracts [52,53,54], as well as from tomatoes [55].

In this study, PEC proved to be a promising antimicrobial agent with antibacterial and antifungal activity on tested microorganisms and major impact against fungal strains (*C. albicans* and *T. rubrum*) and gram-positive bacteria tested. The antimicrobial activity of PEC showed that there was no uniform response among microbial strains, in terms of groups, with some strains being more sensible than others. Interestingly, a selective antimicrobial activity was observed, due to the lack of biocidal effects against human skin, common residents *C. acnes* and *S. epidermidis*.

Moreover, during infections, it has been reported that carotenoids could inhibit the microbial lipopolysaccharides that affect the inflammatory response, reducing the levels of inducible nitric oxide synthase and cyclooxygenase-2 proteins [51]. Although the direct antimicrobial mechanism of PEC is not known, these results establish it as a potential skin antimicrobial agent, as well as a skin prebiotic, due to its selective impact against skin microbiome. Given *C. acnes* and *S. epidermidis* are common human skin-inhabitants, PEC could control the microbiome avoiding the pathogen proliferation, while allowing the growth of normal skin microorganisms.

The potential prebiotic activity was observed on anti-adhesion assay. The microbial attachment to host cells is one of the early strategies for the successful establishment of infections. This process can be mediated by several components, such as adhesins, pili, or fimbriae, as well as specific exopolysaccharides. PEC prevented microbial adhesion to HaCaT monolayers, selectively. *C. acnes* and *S. epidermidis* were able to attach to monolayers after 5 h of incubation; on the contrary, pathogens presented a decrease in its adhesion capacity, progressively. As *C. acnes* and *S. epidermidis* are frequent colonizers of skin and mucosal surfaces, sIC PEC showed a capacity to prevent future infections, while allowing common inhabitants to persist on monolayers. This showed the potential of PEC as a future ingredient for skin care product formulations; however, further studies should clarify the exact structures and mechanisms affected by PEC in the microbial cell, leading to disrupted adhesion.

### 4.3. In Vitro Permeation of PEC Carotenoids

HPLC-PDA-MS/MS identification and quantification also revealed information on the availability of the carotenoids in PEC. From the 20 compounds identified (Appendix A), β-cryptoxanthin and β-carotene, along with their isomeric variants, showed the highest permeation and, therefore, the cyclohexene ring was found to be a determinant in the overall permeability of carotenoids. From a dermatological point of view, it was found that the carotenoids present in PEC behaved according to Fick’s first law on synthetic skin models. The PEC-DMSO and β-carotene-DMSO solutions showed diffusion coefficients (kd) of 2.40 and 0.26, as well as zero and first order kinetics, respectively.

Both the rapid stabilization of carotenoid accumulation and high release rates in the first hours of the carotenoids presented in PEC make theses carotenoids suitable for transdermal delivery, since the protective administration mechanisms for these bioactive compounds are required [56]. The PEC carotenoid results are consistent with those obtained for β-carotene and support those obtained on the HaCaT cell line. The PEC carotenoids are able to penetrate and protect cells against ROS, while the un-permeated fraction would reduce UV-induced skin damage on the surface. Nevertheless, further in vivo and encapsulation assays are required to assess PEC carotenoids’ actual potential.

## 5. Conclusions

The obtained results suggest that PEC has great potential to be used as a functional ingredient in after-sun lotions or creams. PEC carotenoids were internalized by the HaCaT keratinocyte cells and reduced the UV-induced ROS production in treated cells. In this way, PEC exerted a regenerative effect on the previously UV-irradiated HaCaT cells. As expected, the photoprotective and regenerating effects were UV dose-dependent. No cytotoxic effect was observed after the incubation of PEC with HaCaT keratinocytes at the concentration tested. In addition, PEC treatment stimulated the adhesion capacity of the human skin’s common resident microorganisms to HaCaT cells, while displaying a significant anti-adhesive activity against all tested pathogens.

Overall, the data presented here can be considered a starting point for the initiation of the use of PEC as functional ingredient in formulations for photoprotection against solar radiation exposure and its damaging effects on human skin. However, further studies are needed to promote not only the use of PEC in after-sun lotions or creams but also in sunscreens and skin-care cosmetics. Future studies should address, among other issues, the determination of the in vitro sun protection factor (SPF) of PEC, the need for synthetic filters in the sunscreen under development, UV-induced DNA damage, and a better understanding of the observed photoprotective and regenerating effects.

## Figures and Tables

**Figure 1 pharmaceutics-13-01898-f001:**
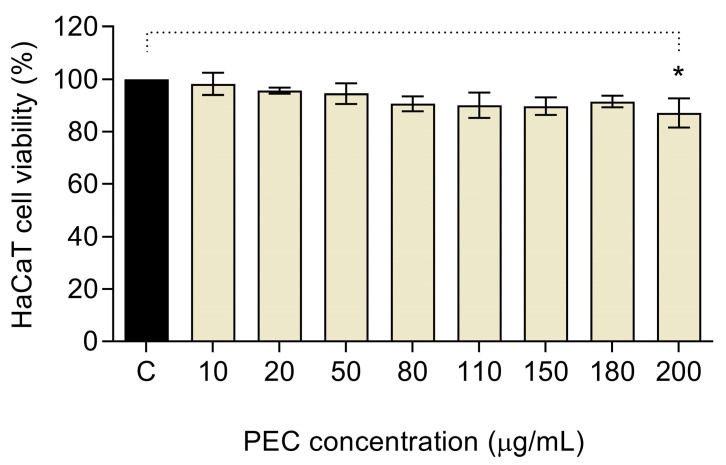
Cytotoxic activities of persimmon byproduct extract (PEC) on human keratinocytes (HaCaT) cell line. PEC treatment at 200 µg/mL provoked a slightly significant reduction on viability (* *p* < 0.05, ns *p* > 0.05, ANOVA, and Dunnett’s post hoc test).

**Figure 2 pharmaceutics-13-01898-f002:**
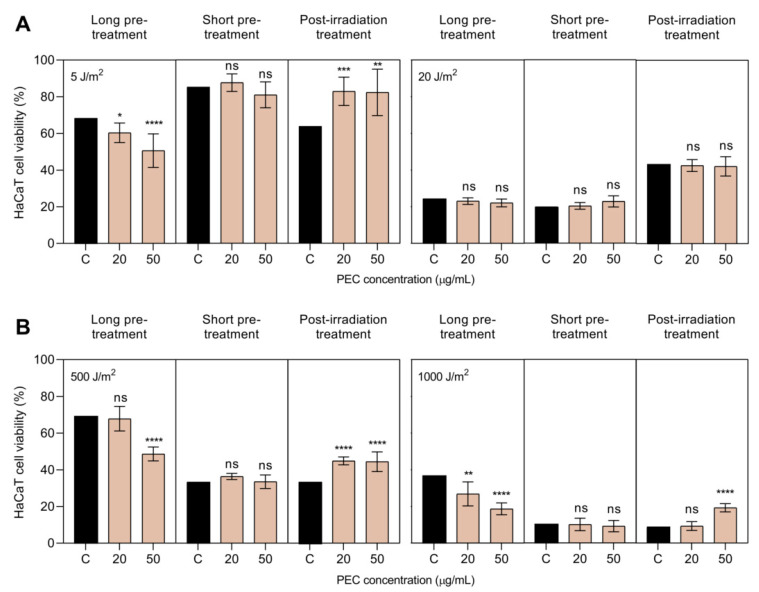
Viability of human keratinocytes (HaCaT) cells after long, short, and post-treatment with persimmon extract rich in carotenoids (PEC) at low (lPEC) and high (hPEC) concentration against (**A**) UV-A (5 and 20 J/m^2^) and (**B**) UV-B (500 and 1000 J/m^2^) irradiation intensities. PEC treatment at lPEC (20 µg/mL) and hPEC (50 µg/mL) led to a significant increase of HaCaT viability post-irradiation, compared to irradiated but non-treated control (C) (**** *p* < 0.0001, *** *p* < 0.001, ** *p* < 0.01, * *p* < 0.05, ns *p* > 0.05, ANOVA, and Dunnett’s post hoc test). All data were normalized, according to a non-irradiated control (100% viability).

**Figure 3 pharmaceutics-13-01898-f003:**
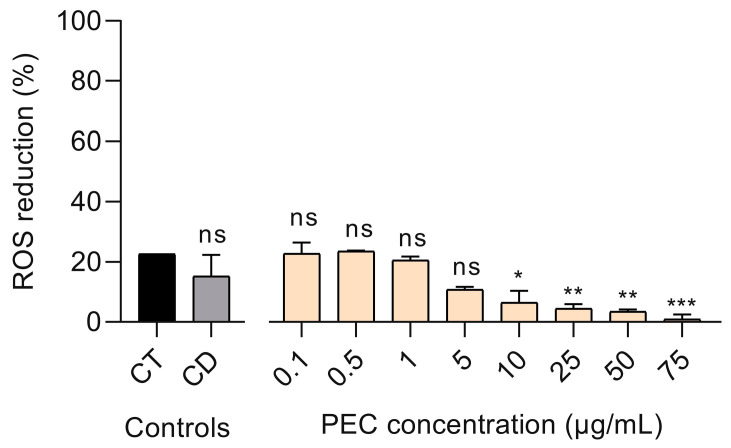
Percentage of reactive oxygen species (ROS) reduction in persimmon extract, rich in carotenoids (PEC)-treated HaCaT cells after UV-B irradiation. PEC treatment from 0.10 to 5.00 µg/mL exhibited a significant reduction in ROS production similarly to Trolox control (CT) (*** *p* < 0.001, ** *p* < 0.01, * *p* < 0.05, ns *p* > 0.05, ANOVA, and Dunnett’s post hoc test). Negative control with DMSO (CD) was used to determine its interference in ROS production. All data were normalized with ROS production in untreated, but irradiated, HaCaT cells.

**Figure 4 pharmaceutics-13-01898-f004:**
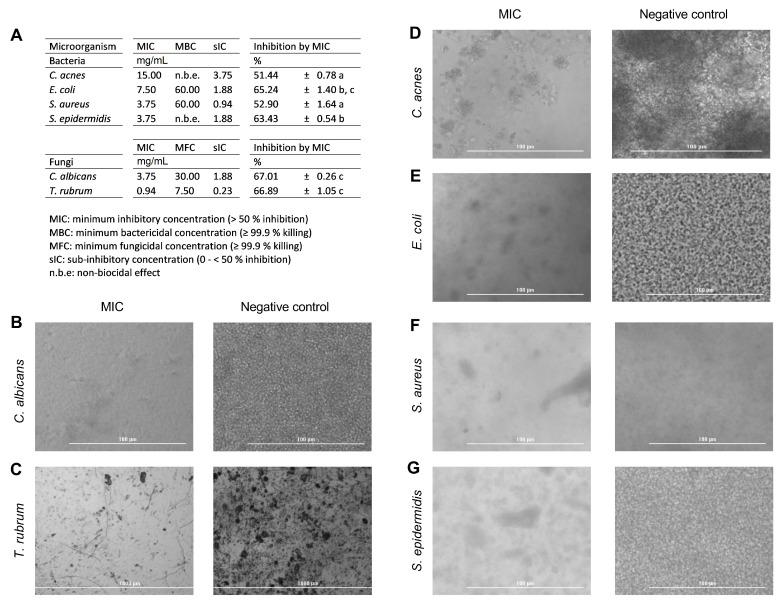
(**A**) Minimum inhibitory (MIC), biocidal (MBC), and sub-inhibitory (sIC) concentrations of persimmon extract rich in carotenoids (PEC) by microdilution assay. MIC values of PEC led to a significant inhibition of microbial strains in comparison with untreated controls (different letters showed significant differences between tested strains, *p* < 0.05, ANOVA and Tukey’s post hoc test). Optical images of wells containing PEC MIC-treated live microorganisms (left) or untreated controls (right): (**B**) *Candida albicans*, (**C**) *Trichophyton rubrum*, (**D**) *Cutibacterium acnes*, (**E**) *Escherichia coli*, (**F**) *Staphylococcus aureus*, and (**G**) *Staphylococcus epidermidis*.

**Figure 5 pharmaceutics-13-01898-f005:**
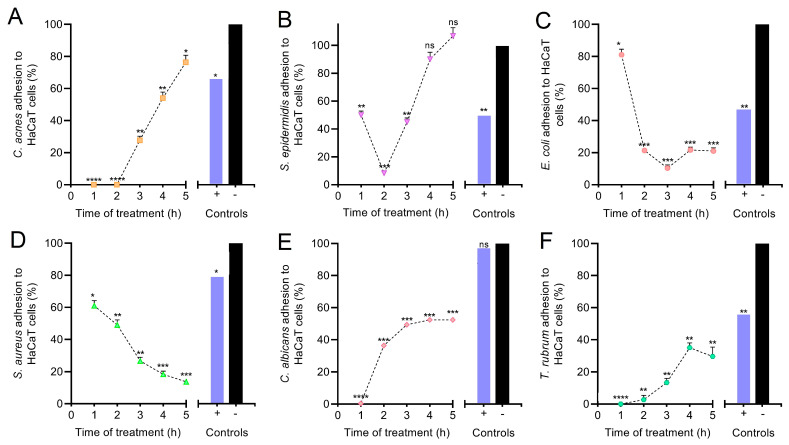
Percentage of microbial adhesion (mean ± SD) to persimmon extract rich in carotenoids (PEC)-treated HaCaT monolayer cells. PEC treatment at MIC/4 led to a significant decrease of microbial adhesion to HaCaT cells over time, as compared to the negative control (**** *p* < 0.0001, *** *p* < 0.001, ** *p* < 0.01, * *p* < 0.05, ns *p* > 0.05, ANOVA, and Dunnett’s post hoc test). Microorganisms: (**A**) *Cutibacterium acnes*, (**B**) *Staphylococcus epidermidis*, (**C**) *Escherichia coli*, (**D**) *Staphylococcus aureus*, (**E**) *Candida albicans*, and (**F**) *Trichophyton rubrum*.

**Figure 6 pharmaceutics-13-01898-f006:**
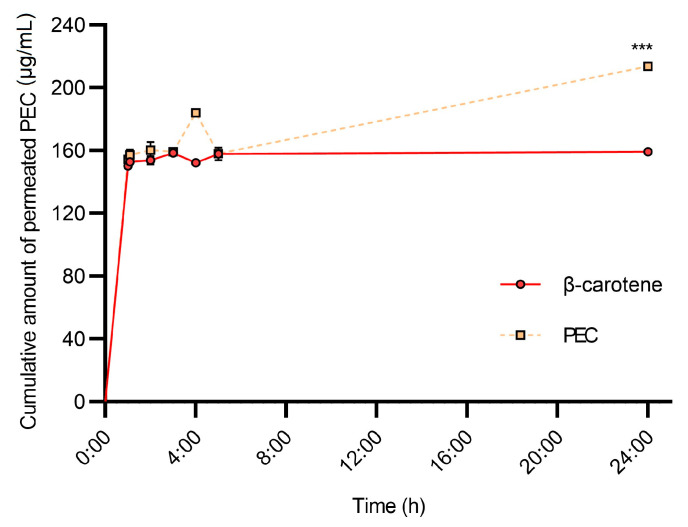
Cumulative amount (µg/mL) of permeated persimmon extract rich in carotenoids (PEC) through Franz diffusion cells synthetic skin membranes model. PEC carotenoids permeated significantly more than the control β-carotene after 24 h of exposure (*** *p* < 0.001, ANOVA, and *t*-test post hoc).

## Data Availability

Data are contained within the article and Appendix A.

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
