# Peer review of "Carotenoids from Persimmon (Diospyros kaki Thunb.) Byproducts Exert Photoprotective, Antioxidative and Microbial Anti-Adhesive Effects on HaCaT"

_pharmaceutics, 2021, doi:10.3390/pharmaceutics13111898_

Round 1

Reviewer 1 Report

  1. Please provide the latin name of the persimmon in the title, abstract and where is for first time mentioned in the body of the manuscript.
  2. Persimmon byproduct (pulp, skin, and seeds) was subjected to a solid-liquid extraction with acetone (1:3 v/v) - please correct it
  3. DMSO could exhibit antimicrobial effects on some microbial strains, so it is not the recommendation to be used as a general negative control - negative control should be the tested strain only diluted in the medium, and the solution with DMSO the confirmation that DMSO no exhibit effects on the tested strains
  4. Which concentration of DMSO for dissolving of tested strains is used?

Reviewer 2 Report

Dear Authors,

your study is very interesting and very well described.

The manuscript is about the potential persimmon byproducts of for use as functional ingredient of skin-care products. It is believed that carotenoids from persimmon fruit processing bioproducts influence on the photoprotective, antioxidative and microbial anti-adhesive effects on HaCaT cells. The results show that the carotenoids extracted from presimmon fruit be used as functional ingredient in after-sun lotions or creams.

Title and abstract are appropriate. Keywords are informative. Introduction perfectly describes the background. Methods and materials are described in detail. Results are well described and discussed. Conclusion responds to the aim of the study.

There are no major recommendations for the improvement of the manuscript in my opinion.

Minor recommendations for the improvement of the manuscript:

  • Please remove method data from the aim of the study eg. Franz diffusion cell system etc.
  • Fig. 1. Can viability be over 100%?
  • Fig. 5B check time unit.
